# Controlling microbial co-culture based on substrate pulsing can lead to stability through differential fitness advantages

J. Andres Martinez[1], Matheo Delvenne[1], Lucas Henrion[1], Fabian Moreno[1], Samuel Telek[1], Christian Dusny[2], Frank Delvigne[1]*

1 TERRA Research and Teaching Centre, Microbial Processes and Interactions (MiPI), Gembloux Agro-Bio Tech, University of Liége, Gembloux, Belgium, 2 Microscale Analysis and Engineering, Department of Solar Materials, Helmholtz-Centre for Environmental Research- UFZ Leipzig, Leipzig, Germany

☯ These authors contributed equally to this work.

* f.delvigne@uliege.be

**Data Availability Statement:** All relevant data are within the manuscript and its Supporting Information files.

## Abstract

Microbial consortia are an exciting alternative for increasing the performances of bioprocesses for the production of complex metabolic products. However, the functional properties of microbial communities remain challenging to control, considering the complex interaction mechanisms occurring between co-cultured microbial species. Indeed, microbial communities are highly dynamic and can adapt to changing environmental conditions through complex mechanisms, such as phenotypic diversification. We focused on stabilizing a co-culture of Saccharomyces cerevisiae and Escherichia coli in continuous cultures. Our preliminary data pointed out that transient diauxic shifts could lead to stable co-culture by providing periodic fitness advantages to the yeast. Based on a computational toolbox called MONCKS (for MONod-type Co-culture Kinetic Simulation), we were able to predict the dynamics of diauxic shift for both species based on a cybernetic approach. This toolbox was further used to predict the frequency of diauxic shift to be applied to reach co-culture stability. These simulations were successfully reproduced experimentally in continuous bioreactors with glucose pulsing. Finally, based on a bet-hedging reporter, we observed that the yeast population exhibited an increased phenotypic diversification process in co-culture compared with mono-culture, suggesting that this mechanism could be the basis of the metabolic fitness of the yeast.

## Author summary

Being able to manipulate the dynamics of microbial co-cultures is a technical challenge that need to be addressed in order to get a deeper insight about how microbial communities are evolving in their ecological context, as well as for exploiting the potential offered by such communities in an applied context e.g., for setting up more robust bioprocesses relying on the use of several microbial species. In this study, we used continuous cultures of bacteria (*E. coli*) and yeast (*S. cerevisiae*) in order to demonstrate that a simple nutrient

**Funding:** This project was supported through different research grants. First, we would like to thank the Era-Cobiotech established based on the H2020 European framework for providing funding (Contibio and ComRaDes projects). Wallonia is also gratefully acknowledged for financial support. This measure is co-financed with tax funds on the basis of the budget passed by the Saxon state parliament. JAM is supported by a post-doctoral grant (Contibio project). FM is supported by a post-doctoral grant (Sunup project, supported by Wallonia). LH is supported by a FRIA PhD grant provided by the Belgian Fund for Scientific Research (FNRS). MD is supported by a PhD grant provided by the Belgian Fund for Scientific Research (FNRS) in the context of an Era-Net Aquatic Pollutant project (ARENA). The funders had no role in study design, data collection and analysis, decision to publish, or preparation of the manuscript.

**Competing interests:** The authors have declared that no competing interests exist.

pulsing strategy can be used for adjusting the composition of the community with time. As expected, during growth on glucose, *E. coli* quickly outcompeted *S. cerevisiae*. However, when glucose is pulsed into the culture, increased metabolic fitness of the yeast was observed upon reconsumption of the main side metabolites i.e., acetate and ethanol, leading to a robust oscillating growth profile for both species. The optimal pulsing frequency was predicted based on a cybernetic version of a Monod growth model taking into account the main metabolic routes involved in the process. Considering the limited number of metabolic details needed, this cybernetic approach could be generalized to other communities.

## Introduction

Microbial communities have colonized every ecosystem on earth, suggesting that access to an almost unlimited biotransformation capability can be granted through the assembly of multiple species within the same cultivation device [1]. Microbial consortia, either natural or synthetic, have therefore been considered a promising platform for the production of diverse metabolites [1–3]. To date, microbial communities have been successfully used for various industrial purposes, such as increasing crop productivity, bioremediation of soils and water bodies, and manufacturing food and pharmaceutical products [1,4–7].

Despite these achievements, the dynamics of microbial consortia remain challenging to control [8], mainly because of the complexity of the mechanisms leading to microbial interactions and their time-dependence on the extracellular conditions [9–12]. In this context, many studies have focused on the use of simplified co-cultures with engineered microbial interactions mechanisms such as programmed quorum sensing [13], the use of synthetic toxin-antitoxin systems [14], or, more commonly, the use of auxotrophic strains [15,16]. More recently, optogenetics, i.e., the use of light to control the activation of light-sensitive gene circuits, has been used to efficiently control microbial co-cultures [17,18]. Despite all these advances, there is still some progress that is needed for the effective manipulation of microbial co-cultures without the use of the genetic toolbox offered by synthetic biology [19]. In this work, we will consider the co-culture of *Escherichia coli* and *Saccharomyces cerevisiae* in continuous culture. These microbes have been selected because of their relevance as model organisms and their widespread use for industrial applications. Since *E. coli* is a very fast grower, it can be expected that the yeast will be outcompeted during the exponential phase of growth. However, these microbes are also able to release side metabolites based on overflow metabolism, i.e., mainly acetate (ACE) and ethanol (ETH), leading to a diauxic shift when the primary carbon source (glucose, GLU) is depleted. This diauxic shift can, in turn, promote the growth of the slow grower yeast, potentially leading to co-culture stability [20]. However, in this case, an essential requirement for stability is that successive diauxic shifts must occur with a given periodicity during the continuous culture through, for example, GLU pulsing. Such periodic perturbation approach has been previously acknowledged as an efficient control strategy for microbial co-cultures [9,11,12], and typically leads to oscillations in cell number, each strain exhibiting period of growth and decay according to their respective metabolic capabilities [21–23]. In order to be able to predict such population behaviour emerging from complex metabolic interactions, mathematical models are needed.

To this end, a Monod-based ODE model was designed in order to capture the main metabolic features of the yeast-bacteria co-culture. The potential fitness advantage offered by diauxic shifts was taken into account based on cybernetic variables reflecting the metabolic

capabilities of each strain in terms of substrate assimilation (in our case GLU, but also ACE and ETH) [24]. This model was used for determining the optimal GLU pulsing frequency leading to population stability, enabling the continuous co-culture of yeast and bacteria. The model was integrated into a modular computational framework called MONCKS (for MONod-type Co-culture Kinetic Simulation), allowing the generalization of the approach to other types of co-cultures. Model predictions were then assessed experimentally in continuous cultures monitored based on online flow cytometry. A fundamental question behind the fitness offered through metabolic flexibility is to what extent this flexibility is due to phenotypic diversification. Indeed, phenotypic diversification has been previously observed for *E. coli*, and *S. cerevisiae* grown under fluctuating environmental conditions [10,25,26], the additional fitness advantage provided by such diversification mechanisms is in accordance with the previously made predictions [21,27]. These diversification mechanisms were experimentally evaluated based on online flow cytometry with and a fluorescent transcriptional reporter, and the possible extension of MONCKS to a stochastic framework was discussed accordingly.

## Materials and methods

### Mathematical modelling: ODEs based on Monod kinetics and cybernetic approach

In this work, we designed a simplified cybernetic mathematical framework for two microbial strains exhibiting different substrate/metabolites consumption profiles and production states [24,28–31]. The basic sets of considered reactions are the assimilation of glucose (GLU) through the oxidative and respiro-fermentative (overflow metabolism) pathways, as well as the assimilation of the metabolites produced through overflow metabolism (i.e., mainly acetate ACE and ethanol ETH in our case). These reactions are modelled based on Monod type equations (*S1 File* and Fig 1).

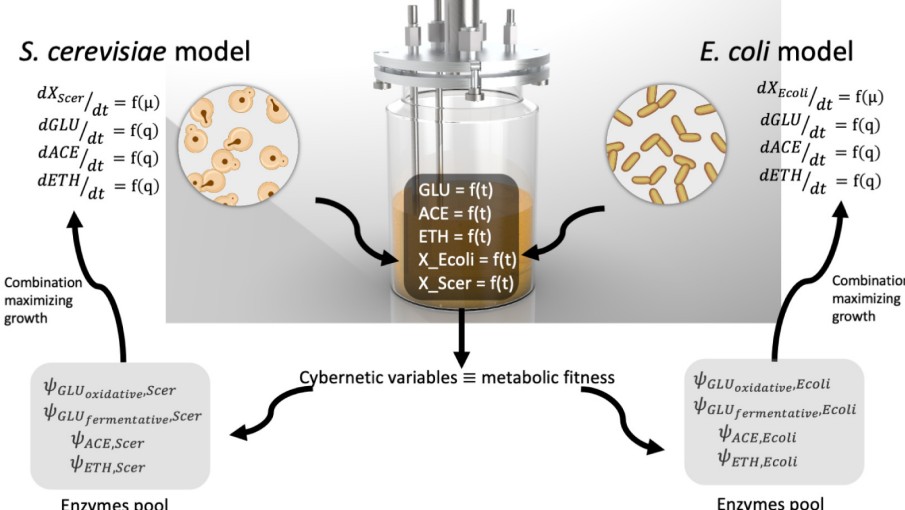

**Fig 1. Simplified representation for the construction of the model for the co-culture experiments.** Each microbial species is taken as a compartment and interacts through the pool of metabolite concentrations. The latter metabolites, along with the biomass concentrations, define the state of the process, while the cybernetic variables that regulate the participation of the different ODEs related to the metabolic pathways in each cell define the metabolic state of each microorganism.

Each microbial species modelled based on a set of ODEs was the and used as a kernel section for a co-culture computational framework. In this framework, the metabolic activities exhibited by each species are optimized based on a cybernetic approach. Briefly, the cybernetic model was based on establishing a common biochemical reaction for any substrate $S$ that the biomass $X$ can consume to grow based on a given metabolic pathway. In the context of the co-cultures between *E. coli* and *S. cerevisiae*, three main substrates $S$ can be considered, i.e., GLU, ACE, and ETH. In the cybernetic approach, the consumption rates of each metabolic pathway are determined by the number of resources allocated by the cell towards synthesizing its enzymatic machinery. The latter is achieved by including an equation accounting for the production of a virtual key enzyme ($\Psi$) required for the assimilation of the main carbon sources. In this work, the pathways related to substrate consumption can be written as:

$$X + Y_{\frac{s}{\sigma}}S \xrightarrow{\Psi_\sigma} X + Y_{\frac{x}{\sigma}}X + Y_{\frac{a}{\sigma}}A + \cdots + Y_{\frac{n}{\sigma}}N \qquad \text{Eq 1}$$

$$X \xrightarrow{S} \Psi_\sigma + X' \qquad \text{Eq 2}$$

where $X'$ is the biomass excluding $\Psi_\sigma$ while $A$ to $N$ are byproducts produced at their Yields *(Y)*. Therefore, a stoichiometric vector $\psi_\sigma$ for any component in the media $M_i$ (where $i = X$, $S$, $A, \ldots, N)$ can be constructed given the consumption of $S$. The rate of change can therefore be defined by the product of $\psi_\sigma$ and the $S$ consumption rate $r_\sigma$ at the given $\Psi_\sigma$ concentration, such that:

$$\frac{dM}{dt} = \psi_\sigma r_\sigma = \begin{bmatrix} Y_{\frac{x}{\sigma}} \\ Y_{\frac{s}{\sigma}} \\ Y_{\frac{a}{\sigma}} \\ \vdots \\ Y_{\frac{n}{\sigma}} \end{bmatrix} \frac{q_\sigma \Psi_\sigma S}{K_\sigma + S} X \qquad \text{Eq 3}$$

and the rate of production of $\Psi_\sigma$ can be described by the following equation:

$$\frac{d\Psi_\sigma}{dt} = \varepsilon^c + \varepsilon^i \frac{SX}{K'_\sigma + S} - \delta\Psi_\sigma - \mu\Psi_\sigma \qquad \text{Eq 4}$$

where $\varepsilon^c$ and $\varepsilon^i$ are the constitutive and inducible expression rates, respectively. $\delta$ is the degradation rate, and $\mu$ is the growth rate. The value of $q_\sigma\Psi_\sigma$ is maximized when the organism has committed to the investment of the maximum amount of resources towards this metabolic path. Therefore, for the simulations, the value of is not needed to be precisely known, but the relative value [24,28–31], given that:

$$q_\sigma\Psi_\sigma = q_\sigma^{max}\left[\frac{\Psi_\sigma}{\Psi_\sigma^{max}}\right] = q_\sigma^{max}\Psi_\sigma^{rel} \qquad \text{Eq 5}$$

where $\Psi_\sigma{}^{max}$ can be defined by the terms of the maximum production, degradation, and

dilution rates from Eq 4. Using Eq 5 on Eq 3 and reordering, we obtain:

$$
\frac{dM}{dt} = \begin{bmatrix} \dfrac{Y_x}{\sigma} \\[6pt] \dfrac{Y_s}{\sigma} \\[6pt] \dfrac{Y_a}{\sigma} \\[6pt] \vdots \\[6pt] \dfrac{Y_n}{\sigma} \end{bmatrix} q_\sigma^{max} \Psi_\sigma^{rel} \frac{S}{K_\sigma + S} X = \psi_\sigma q_\sigma^{max} \, \Psi_\sigma^{rel} \, H_\sigma X
\qquad \text{Eq 6}
$$

where $H_\sigma$ is a Monod-type hill function for uptake rate reduction at low substrate concentrations. The later Hill function can be modified to address other types of substrate and metabolites uptake rate interactions (e.g., product inhibition, competitive inhibition, see **S1 File**). Eq 6 can be extended for several known metabolic pathways (E.g., oxidative and fermentative glucose consumption, re-consumption of byproducts). By extending the matrix and vector dimensions as follows:

$$
\frac{dM}{dt} = \begin{bmatrix} \dfrac{Y_x}{\sigma} & \dfrac{Y_x}{\alpha} & \cdots & \dfrac{Y_x}{\eta} \\[6pt] \dfrac{Y_s}{\sigma} & \dfrac{Y_s}{\alpha} & \cdots & \dfrac{Y_s}{\eta} \\[6pt] \dfrac{Y_a}{\sigma} & \dfrac{Y_a}{\alpha} & \cdots & \dfrac{Y_a}{\eta} \\[6pt] \vdots & \vdots & \ddots & \vdots \\[6pt] \dfrac{Y_n}{\sigma} & \dfrac{Y_n}{\alpha} & \cdots & \dfrac{Y_n}{\eta} \end{bmatrix} \begin{bmatrix} q_\sigma^{max} \, \Psi_\sigma^{rel} \, H_\sigma X \\[6pt] q_\alpha^{max} \, \Psi_\alpha^{rel} \, H_\alpha X \\[6pt] \vdots \\[6pt] q_\eta^{max} \, \Psi_\eta^{rel} \, H_\eta X \end{bmatrix}
\qquad \text{Eq 7}
$$

where each column represents the stoichiometric behaviour associated with a specific metabolic pathway. The global metabolic state ($\Phi$) of the cell is given by the linear combination of all the activities of the individual metabolic pathways on the metabolite pool $M$. Finally, to better accomplish the latter, the cybernetic modeling approach introduces the regulation to the virtual enzymes ($\Psi^{\sigma \cdots \eta}{}_{rel}$) expression and activity. This regulation is achieved by multiplying the rates of enzyme synthesis and activity by the cybernetic variables $\upsilon$ and $\nu$ calculated by matching law equations constructed for specific metabolic objectives. In this work, the growth rate was used as a metabolic objective as follows:

$$
\upsilon_i = \frac{\mu_i}{\sum_{j=1}^{\omega} \mu_j}
\qquad \text{Eq 8}
$$

$$
\nu_i = \frac{\mu_i}{max(\mu_{1\ldots\omega})}
\qquad \text{Eq 9}
$$

In this way, the physiological behaviour (*P*) on the cultivation conditions could be understood as a function of the metabolic state and the environment:

$$
P(\Phi, M) \approx \langle \Psi | M \rangle \approx \langle \Sigma_\sigma^\eta \Psi_{(\upsilon,\nu)} | M \rangle
\qquad \text{Eq 10}
$$

Interestingly, this function is approximated by the contributions of the organism's specific enzyme content operating on the environmental conditions $M$ (metabolite concentrations, in

this work). The latter allows to tie the metabolic behavior and regulation of the cell to fitness if the performance is attached to the organism's survival.

## Computational toolbox for the simulation of continuous co-cultures

The models obtained for each microbial species were combined for running co-culture simulations in continuous bioreactors. Three different parameters were considered, Dilution rate (*D*), frequency of pulsing (*w*), and time fraction for feed pulse (*s*). Pulses were applied to the dilution rate resulting in discontinuous feeding media with 30 g/L of GLU as carbon source. The feeding regimes were square wave functions where the s determines the symmetry of the pulse wave. Substrate consumption and biomass/metabolite yield parameters for each strain were approximated from batch and chemostat experiments. Variables for enzymatic production and degradation were set according to published cybernetic models for *E. coli* and *S. cerevisiae* [24,28] (*S1 File*). Simulations were performed with the MATLAB-based Monod-type Co-culture Kinetic Simulation (MONCKS) toolbox that allows using two cybernetic models simultaneously integrated into a bioreactor external metabolite concentration simulation. Briefly, the toolbox allows the models designed for each microbial species to interact with the environmental variables (i.e., GLU, ACE, ETH) at each integration time step. These interactions occur at equal probability for each species. This toolbox is available at https://gitlab.uliege.be/mipi/published-software/mbms-toolbox, used databases can be found at https://gitlab.uliege.be/mipi/published-software/mipi-model-and-simulation-database and further description and a simplified simulation workflow can be found in the supporting information (*S2 File*).

## Strains and medium composition

Strains used in this study were the *Escherichia coli* K-12 W3110, *Saccharomyces cerevisiae* CEN-PK 117D, and a GFP expressing *Saccharomyces cerevisiae* strain Pglc3::GFP which grow on Verduyn minimal media. This strain is derived from the CEN-PK 117D background expressing GFP under the control of a chimeric promoter which upregulates the expression of GFP under nutrient-limiting conditions [32,33]. The strains were maintained at -80°C in working vials (2 mL) in LB with 30% glycerol (w/v). Precultures and cultures were performed on synthetic media according to Verduyn et al. [34], but with modified phosphate buffer proportions of potassium dihydrogen phosphate (6.309 g/L) and potassium hydrogen phosphate (9.34 g/L) at pH 6.8 and supplemented with various glucose concentrations (Sigma-Aldrich, US).

## Cultivation tests for determining model parameters

The parameters used for modelling microbial strain dynamics were acquired based on two sets of experiments: i) Mini-bioreactor batch experiments involving the strains cultivated separately at different GLU; ii) Chemostat experiments at three different dilution rates (*D*). The mini-bioreactor experiments were performed on a BioLector/RoboLector (M2PLabs, Germany). Six initial GLU concentrations were set in 48-deepwells microplates with values ranging between 20 g/L and 1.25 g/L. Process conditions were set to be constant pH 6.8 and a shaking frequency of 1000 rpm; total bioprocess time was 30 hours for *E. coli* experiments and 36 hours for *S. cerevisiae*. The initial biomass concentration was approximately 0.1 OD for all experiments. Eight wells were used as a set of parallel fermentations for each condition and sequentially taken as a sample every 3 hours. Samples were separated into two vials, one was used for flow cytometry analysis, and the other was immediately filtered and prepared for measuring glucose and organic acids by HPLC.

Chemostat experiments were performed in Dasgip bioreactors with 150 mL of working volume at constant pH 6.8, 1000 rpm, and 30°C. Three dilution rates, i.e., 0.1, 0.2, and 0.3 h$^{-1}$,

were sequentially imposed on the system, with at least five retention times between each step change. Three samples were taken at the last retention time and analysed by spectrometry at OD 600, flow cytometry (FC), and HPLC to account for biomass, GLU, ACE, ETH, GFP fluorescence, and biomass size distribution and concentration by FC analysis. Dasgip experiments were done by triplicate for the *E. coli* and the *S. cerevisiae* Pglc3::GFP strains.

## Co-culture experiments in continuous bioreactors

Co-culture experiments were performed in lab-scale stirred bioreactors (Biostat B-Twin, Sartorius). The processes were performed with a working volume of 1L at a constant temperature of 30°C, pH at 6.8, stirring rate of 1000 rpm, and aeration rate of 1 VVM. Cultures were started with a population ratio of 1:1 *S. cerevisiae*: *E. coli* in g/L units) in the Verduyn modified media. Every experiment batch phase was followed before the feed was started after 10 hours of cultivation. A dilution rate of 0.1 $h^{-1}$ was considered for the chemostat and the GLU pulsed continuous cultivations. For the pulsed cultivations, the pulsing phases were applied as square waves either at a low-frequency ($w = 0.14$ and $s = 0.28$) or a hi-frequency ($w = 0.33$, $s = 0.33$) pulsing regime, resulting in feeds pulses of 2h:5h and 1h:2h (on:off at $D = 0.1$), respectively. All co-culture experiments lasted at least 80 hours, and all were performed at least in duplicate.

## Sample processing

**HPLC.** Samples from fermentation experiments were processed for glucose and organic acid measurement by HPLC with an Aminex HPX-87H column (Bio-Rad, Hercules CA, USA) at 45°C and 5 mM Sulfuric acid as mobile phase. An Agilent 1200 Series HPLC system was used with a refraction index detector at 50°C (Agilent, Santa Clara, CA, USA).

**Flow cytometry.** Flow cytometry data were obtained with an Accuri C6 flow cytometer (BD Accuri, San Jose, CA, USA). The sample was first tested on the C6 FC to measure the events/L and prepared by dilution until a concentration below 1000. Diluted samples are then fed into the C6 FC for analysis at an average flow rate of 14 µl/min with a threshold FSC-H set at 40,000. The analysis ended after collecting at least 40,000 events or 70 µL of the sample. Samples for Online Flow Cytometry were taken with the help of the online Segregostat device, which allows for sampling acquisition, dilution, measurement and control on continuous fermentation processes [35,36].

Flow cytometry data was cleaned from electrical noise and doublets before further analysis. Doublet treatment was performed by linear regression between the area and height of the front scattering signal and eliminating data points based on Pearson standardized residual values above 2. All samples with more than 5% of doublets and or with total remaining events of less than 20,000 were not considered for further analysis. *E. coli* and *S. cerevisiae* were clustered by simple gating based on the forward scattering area signal (FSC-A) (*S3 File*). For GFP fluorescent measurements, gating was also used. Calculations of fractions of events relative to each strain and their fluorescence were performed with the MiPI Flow Cytometry Analysis toolbox (mFCAtoolbox) available at https://gitlab.uliege.be/mipi/published-software/mfca-toolbox, and further description and a simplified analysis workflow can be found in the supporting information (*S3 File*).

## Microfluidic cultivation and time-lapse microscopy

Cells have been cultivated in microfluidic chips provided by Alexander Grünberger's lab (ref. 24W, chambers size: 80 µm x 80 µm x ~850 nm) [37], in Verduyn medium with different glucose concentrations (5 µM, 0.1 mM, 0.2 mM, 0.4 mM, 0.6 mM, 1 mM and 3 mM). The temperature was set at 30°C. The chambers were inoculated with one or two cells by flushing the

device with a cell suspension (OD600 between 0.4 and 0.5). At least 6 cultivation chambers were selected manually for each glucose concentration condition. Microscopy images were acquired using a Nikon Eclipse Ti2-E inverted automated epifluorescence microscope (Nikon Eclipse Ti2-E, Nikon France, France) equipped with a DS-Qi2 camera (Nikon camera DSQi2, Nikon France, France), a 100× oil objective (CFI P-Apo DM Lambda 100× Oil (Ph3), Nikon France, France). The GFP-3035D cube (excitation filter: 472/30 nm, dichroic mirror: 495 nm, emission filter: 520/35 nm, Nikon France, Nikon) was used to measure GFP. The phase contrast images were recorded with an exposure time of 300 ms and an illuminator's intensity of 30%. The GFP images were recorded with an exposure time of 500 ms and an illuminator's intensity of 2% (SOLA SE II, Lumencor, USA). During the first 48 hour, GFP and phase contrast images were acquired every hour. During the 24 last hours, phase contrast images were acquired every 6 minute and GFP images every hour. The optical parameters and the time-lapse were managed with the NIS-Elements Imaging Software (Nikon NIS Elements AR software package, Nikon France, France).

### Growth Rate measurement

The instantaneous growth rate has been approximated by the one of the areas of the colonies in the chambers. Using a homemade Python code, the areas of the colonies were measured and the instantaneous growth rate has been computed (instantaneous $\mu = (\text{area}_{ti} - \text{area}_{ti-1})/ \text{area}_{ti-1}$ for at least three chambers per glucose concentration condition, during the firsts hours until a cell go out of the chamber.

### GFP positive fraction

The fraction of GFP positive cells in the chambers were computed for the 24 last hours of the time-lapse for at least 3 chambers per glucose concentration condition. The cell-segmentation of the images and the measure of single-cell mean GFP intensity were performed using the Python GUI of YeaZ [38]. Cells have been considered as GFP positive when their mean GFP intensity was above 22.5 arbitrary fluorescence units.

## Results

### Cybernetic modelling reveals differences in metabolic fitness upon the diauxic shift in mono-cultures

This first section will be focused i) on the calibration of the cybernetic model and ii) on the characterization of the metabolic phenotypes exhibited by *E. coli* and *S. cerevisiae* when grown separately. For this purpose, we have designed a modular simulation toolbox called MONCKS, each module being one microbial species involved in the co-culture. Based on this toolbox, the growth of each strain will be modelled as a set of four ODEs representing the time trajectories of the biomass and the three main substrates (GLU, ACE, ETH). These three substrates have been selected as they are the most relevant compounds found during the cultivation of each strain on a minimal medium with GLU as the main carbon source. ACE and ETH are typically produced by overflow metabolism and can be further re-consumed as a carbon source when GLU is limiting, leading to diauxic behaviour. During their growth under these conditions, the two microbial strains under consideration will have then the possibility to grow on three different carbon sources. The order of assimilation and the resulting substrate consumption rate will be modelled based on a cybernetic framework. In short, this cybernetic framework allows cells to make the best decision, in terms of substrates consumption, for optimizing their growth rate (**Fig 1**). Before going more into the details of this cybernetic approach, we will

focus first on the determination of the basic growth parameters for each strain grown separately. For this purpose, batch experiments were conducted on minimal medium, and the four state variables (biomass, ETH, GLU, and ACE) were followed in function of time (**Fig 2**).

A diauxic effect was observed for the two strains when grown with high GLU concentration but was more pronounced for *S. cerevisiae*. This trend can be explained by the fact that *S. cerevisiae* is known to favour respiro-fermentative metabolism when growing at near maximum growth rate or under dissolved oxygen-limited conditions [39]. Interestingly, the diauxic shift has been recently recognized as an efficient driver for promoting community stability [20]. The data obtained was then further used for parameter estimation (**Table 1**).

Based on these parameters, it can be concluded that *S. cerevisiae* exhibits generally higher biomass yields ($Y_{x/s}$) and smaller saturation constants ($K_s$), while *E. coli* generally exhibits higher growth rates ($\mu_{max}$) and substrate consumption rates ($q_s^{max}$). Accordingly, it can be concluded that *E. coli* will outgrow *S. cerevisiae* in most cases. However, a window of opportunity leading to potential stable co-culture is offered by the higher affinity of *S. cerevisiae* for low substrate concentration and by its higher metabolic flexibility (diauxic shift) for the consumption of alternative carbon sources, such as ACE and ETH. This feature is particularly interesting since it is known that metabolic flexibility can promote relevant microbial interactions needed for sustaining the establishment of robust microbial communities [23,40].

Ground-breaking theoretical works established that cell-to-cell differences in gene expression can split a population of cells into two or more subpopulations exhibiting different phenotypic/metabolic functions [21,27]. More importantly, it has been suggested that cells are able to switch between these different phenotypic states according to a rate correlated to the environmental changes [35,36,41]. Therefore, it should be possible to modify the distribution of cells into different phenotypic/metabolic states based on controlled environmental fluctuations. However, before being able to verify this hypothesis, we need to incorporate the possibility for the cell to occupy different phenotypic/metabolic states into the model. The latter is exactly the purpose of the cybernetic approach that will be used in this work in order to find environmental conditions leading to co-culture stability. Briefly, the cybernetic algorithm allows for the determination of cybernetic variables corresponding to the main metabolic pathways used by the microbial strains for consuming the different substrates. These variables regulate the quantity of key enzymes ($\Psi$) used for the assimilation of the main substrate (GLU) and the alternative carbon sources (ACE and ETH) (**Fig 1**). Please note that GLU can be assimilated based on the oxidative or the fermentative pathway. Via these variables, the different substrate assimilation quantities are dynamically adjusted for the strains to optimize the growth rate in function of the environmental variables. These cybernetic model parameters were then determined based on the dataset gathered from mono-culture experiments (**Fig 3**), confirming the fact that *E. coli* exhibits less metabolic flexibility/fitness than *S. cerevisiae*.

Indeed, *E. coli* exhibits a clear diauxic profile from GLU to ACE (**Fig 3A**). The cybernetic variable associated with glucose fermentation starts at its maximum value readily after the beginning of the culture and remains as such if the GLU is non-limiting (**Fig 3A and 3B**). Between the range of initial concentration tested (20 to 1.25 g/L), the maximum growth rate achieved by *E. coli* is reduced by only 12.4% (**Fig 3B**), while the model $K_s$ was found to be near 0.11 g/L, where we expect a 50% growth rate decrease. The reduction observed in Fig 3 B on growth rate was found on times with GLU at least below to approx. 0.5g/L, where this substrate limitation becomes stringent (Figs 2 and 3B). Upon GLU limitation, the cybernetic variable associated with ACE consumption increases and peaks at approximately 17 hours. However, the contribution of this cybernetic variable to the global growth of *E. coli* is quite low compared to the one provided by GLU consumption (**Fig 3A and 3C**). On the other hand, *S. cerevisiae* displays higher metabolic flexibility, given the fact that ETH consumption occurs even before

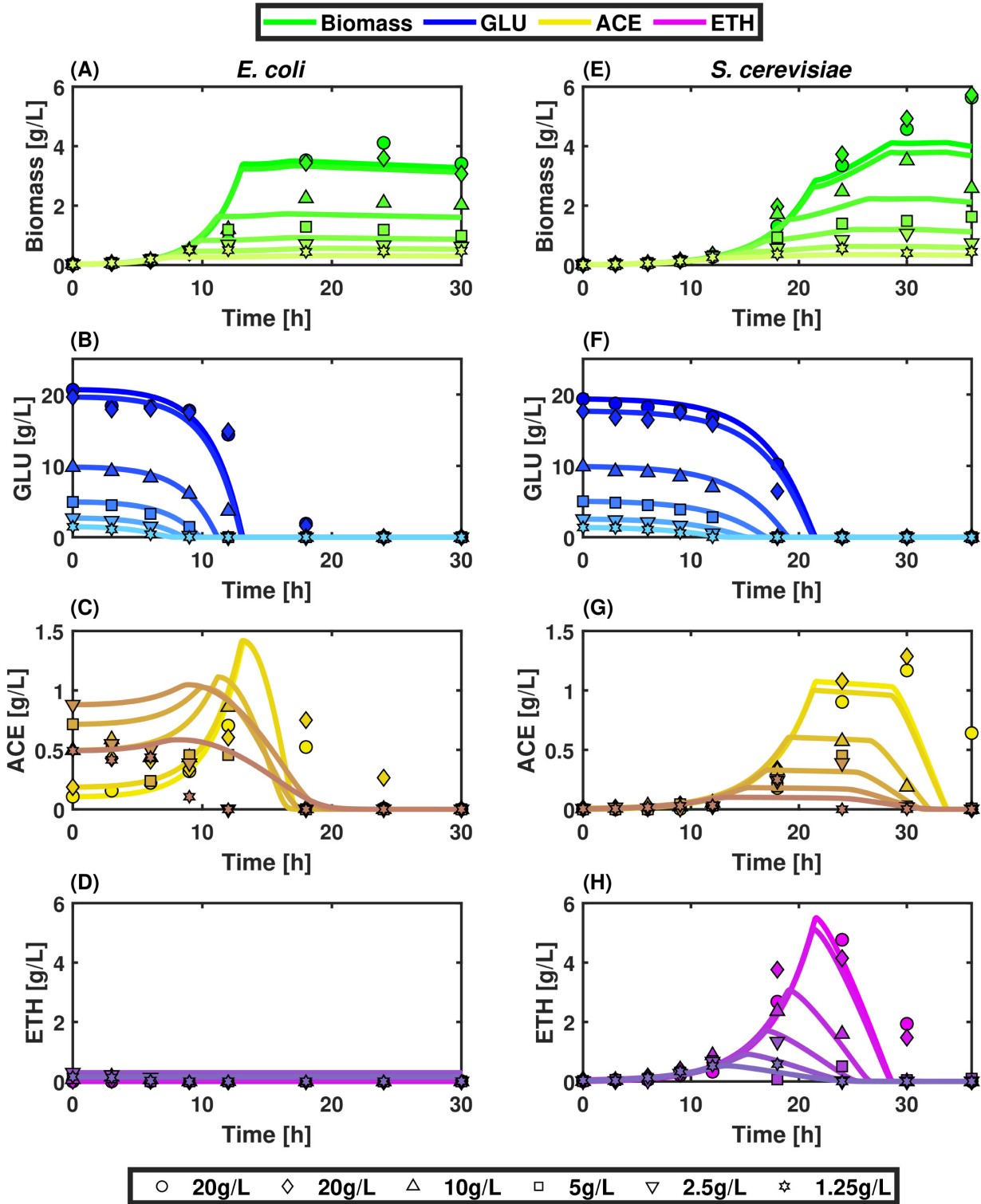

**Fig 2. Biomass growth and metabolite profiles for E. coli and S. cerevisiae grown in microplate Mini-bioreactors at different initial glucose concentrations ([GLU]$_i$).** Experimental data set concentrations shown are: 20 g/L, 10 g/L, 5 g/L, 2.5 g/L and 1.25 g/L. Lines represent model results for Biomass in green, glucose in blue, acetate in yellow, and ethanol in purple.

**Table 1. Biomass maximum growth rates, saturation constants, maximum substrate consumption rates and biomass/substrate yields obtained from modelling approximation to mono-culture fermentations.**

| Substrate | $\mu_{max}$ | | $K_s$ | | $q^s_{max}$ | | $Y_{x/s}$ | |
|---|---|---|---|---|---|---|---|---|
| | *E. coli S. cer* | | *E. coli S. cer* | | *E. coli S. cer* | | *E. coli S. cer* | |
| GLU (ferm) | 0.437 0.299 | | 0.112 0.082 | | -2.640–1.648 | | 0.165 0.182 | |
| ACE | 0.077 0.017 | | 0.108 0.007 | | -0.630–0.058 | | 0.122 0.285 | |
| ETH | ——— 0.080 | | ——— 0.055 | | ——— -0.276 | | ——— 0.287 | |
| GLU(Ox) | 0.234 0.221 | | 1.000 1.469 | | -1.745–3.175 | | 0.134 0.070 | |

the exhaustion of GLU, probably due to a weaker catabolite repression effect (**Fig 3B**). The time profiles of the cybernetic variables show faster accumulation and decay, suggesting higher flexibility for the temporal allocation of resources (**Fig 3B and 3D**). The latter results in a more efficient growth under GLU limiting conditions and on the overflow metabolites ACE and ETH. These observations altogether point out that the metabolic flexibility of *S. cerevisiae* could be exploited to provide a fitness advantage to this strain through the manipulation of the environmental variables, further ensuring its co-cultivation with *E. coli*.

## Fitness disparity between metabolic phenotypes leads to population imbalances during continuous cultures with constant environmental conditions

Continuous cultivation is a mode of operation known to promote the appearance of cells exhibiting different metabolic states [42]. Since we want to challenge the fact that stable co-culture can be obtained based on the active metabolic diversification of microbial species, continuous cultivation has been selected as a relevant device for our experiments. *E. coli* and *S. cerevisiae* were then co-cultivated in continuous bioreactors ($D = 0.1$ h$^{-1}$), and the cultures were monitored by on-line flow cytometry (**Fig 4A**) and sampled for metabolites quantification (*S4 File*).

The evolution of the relative FC events fraction indicates that *E. coli* rose rapidly during the first cultivation phase (from 0 to 10 hours) and outcompeted *S. cerevisiae*. Upon GLU exhaustion (after approx. 10 hours of cultivation), the fraction of *S. cerevisiae* increased based on the consumption of the ACE and ETH released during the initial phase of the culture. However, this effect was only transient, and *S. cerevisiae* was outcompeted again upon the exhaustion of the overflow metabolites (after approx. 40 hours), impairing the stability of the co-culture. The cybernetic model was also used for simulating the time trajectories of the different cybernetic variables for *E. coli* (**Fig 4B**) and *S. cerevisiae* (**Fig 4C**), and the values obtained reflect the higher metabolic fitness of the yeast upon by-products reassimilation (ETH and ACE). Stable co-culture is an essential requirement for the efficient exploitation of microbial resources through, for example, advanced bioprocessing [40,43,44]. Besides the tools offered by synthetic biology, there is actually a lack of efficient actuators for ensuring microbial stability in continuous bioreactors [45]. Interestingly, these data suggest that the transient effect observed in a basic chemostat setup could be extended and maintained in continuous cultures based on GLU pulsing (**Fig 5**). This specific feature will be challenged in the next section based on the cybernetic modelling framework developed in this work.

## Fluctuating environmental conditions provide periodic fitness advantage and can lead to population stability in continuous cultures

We previously observed a transient diauxic effect promoting co-culture stability in continuous culture, and the idea is to extend this effect based on GLU pulsing. We then challenged this

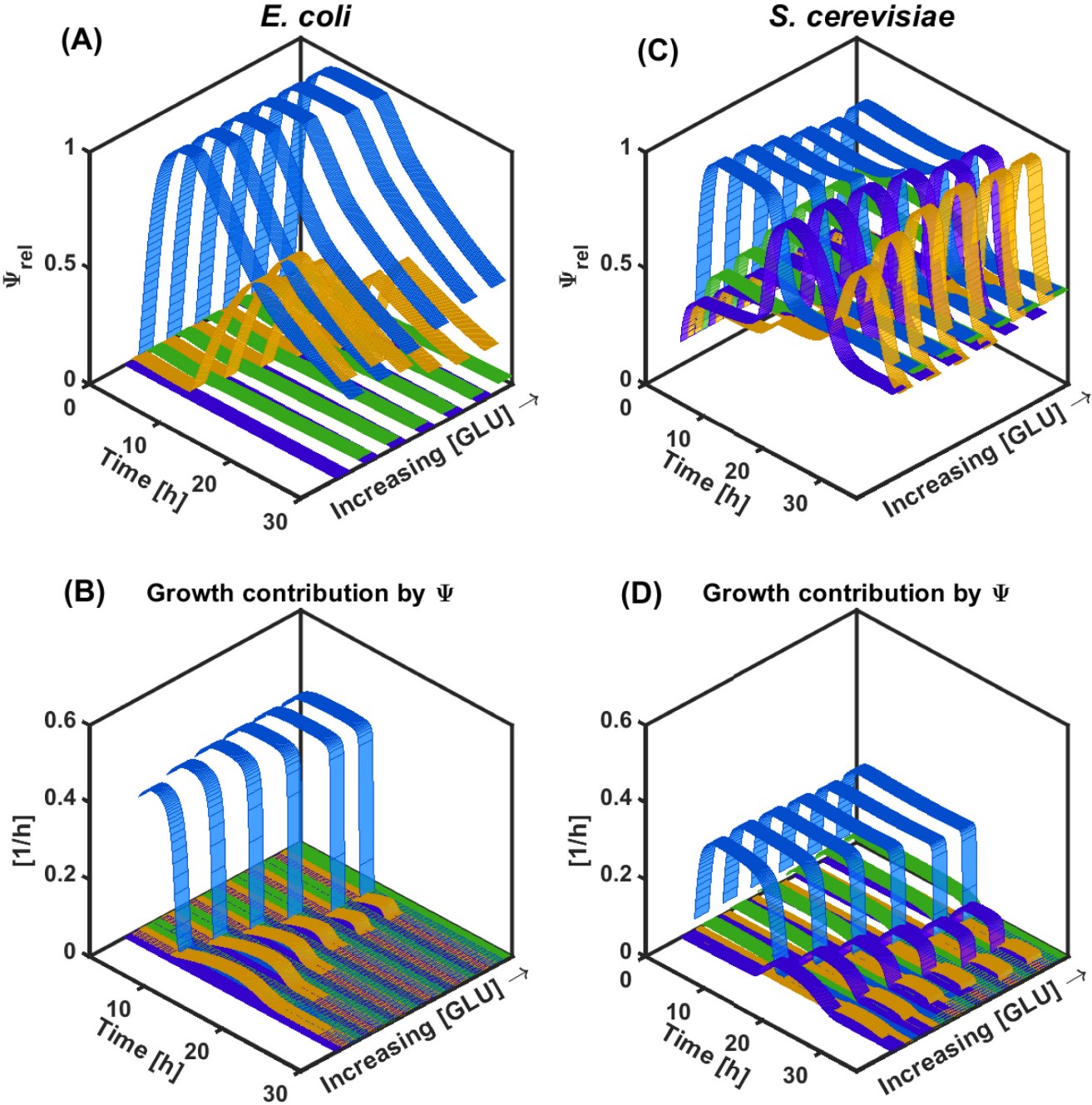

**Fig 3.** Time evolution of $\Psi_{rel}$ for each metabolic pathway at different starting GLU concentrations for E. coli (A) and S. cerevisiae (C). Time evolution of the corresponding contribution of the different metabolic pathways to the growth rate for E. coli (B) and S. cerevisiae (D). Metabolic pathways are colour coded as follows: GLU fermentative/unbalanced consumption in light blue, ACE consumption in yellow, ETH consumption in purple, and GLU oxidative/balanced consumption in green.

hypothesis based on our cybernetic modelling framework. The goal here is to extend the transient fitness advantage provided by the diauxic shift by pulsing glucose according to specific frequencies and amplitudes during continuous cultivation (**Fig 5**). Pulse profiles were defined by square waves with dilution rates ($D$), frequencies ($w$) and pulse duration fraction ($s$).

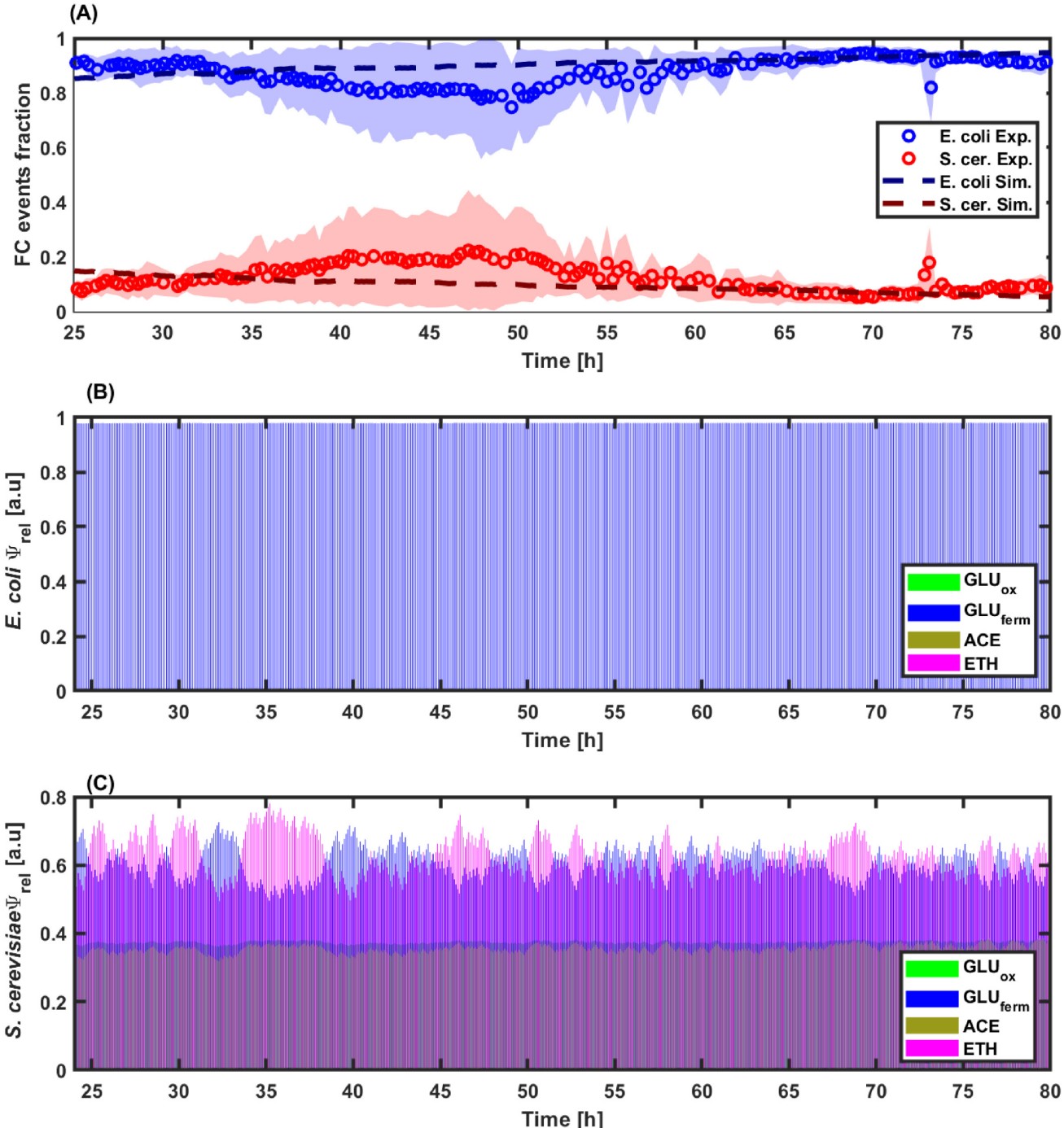

**Fig 4.** A) Time evolution of the FC events fraction of E. coli (blue) and S. cerevisiae (red) during continuous co-culture at a dilution rate of 0.1 h⁻¹. Experimental data are represented by markers and simulation data by dashed lines. B) and C) Time evolution of the relative enzyme ($\Psi_{rel}$) corresponding to each metabolic pathway for E. coli and S. cerevisiae, respectively. Metabolic paths are colour coded as follows: GLU fermentative/unbalanced consumption in blue, ACE consumption in yellow, ETH consumption in purple, and GLU oxidative/balanced consumption in green.

It has indeed been previously shown that environmental perturbations can promote stability in microbial communities [9,11,12,46]. The search for optimal stabilization conditions is a hot topic in the field of microbial ecology [47,48]. We then simulated different co-culture

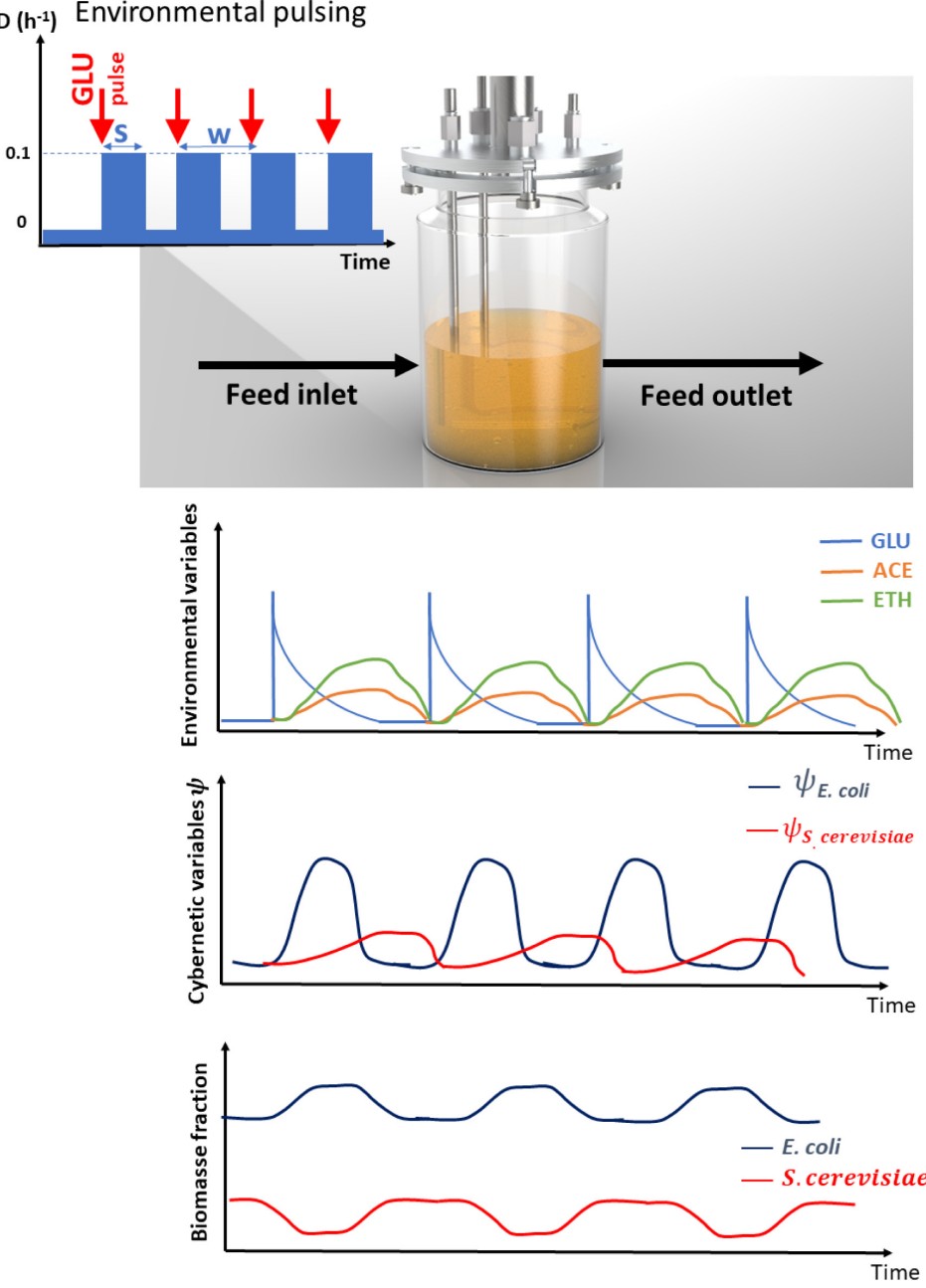

**Fig 5. Simplified representation of the MONCKS simulation framework used for various intermittent feeding profiles.** The feeding profiles are considered as input square waves with dilution rate (D), (w) as the frequency, and (s) as the pulse duration. Model outputs comprised the metabolite availability in the system. The environmental conditions will impact the transitions of $\Psi_{rel}$ for both competing strains. The latter would create a temporary fitness advantage for each strain, which is expected to stabilize the biomass fraction through fermentation. S. cerevisiae in red and E. coli in dark blue.

scenarios in continuous systems for a range of dilution rates (*D*, between 0.025 h$^{-1}$ and 0.3 h$^{-1}$) and for different GLU pulsing parameters (frequencies *w* and step times *s*). Based on the cybernetic model, we simulated the time evolution of the relative fraction of *E. coli* and *S. cerevisiae*

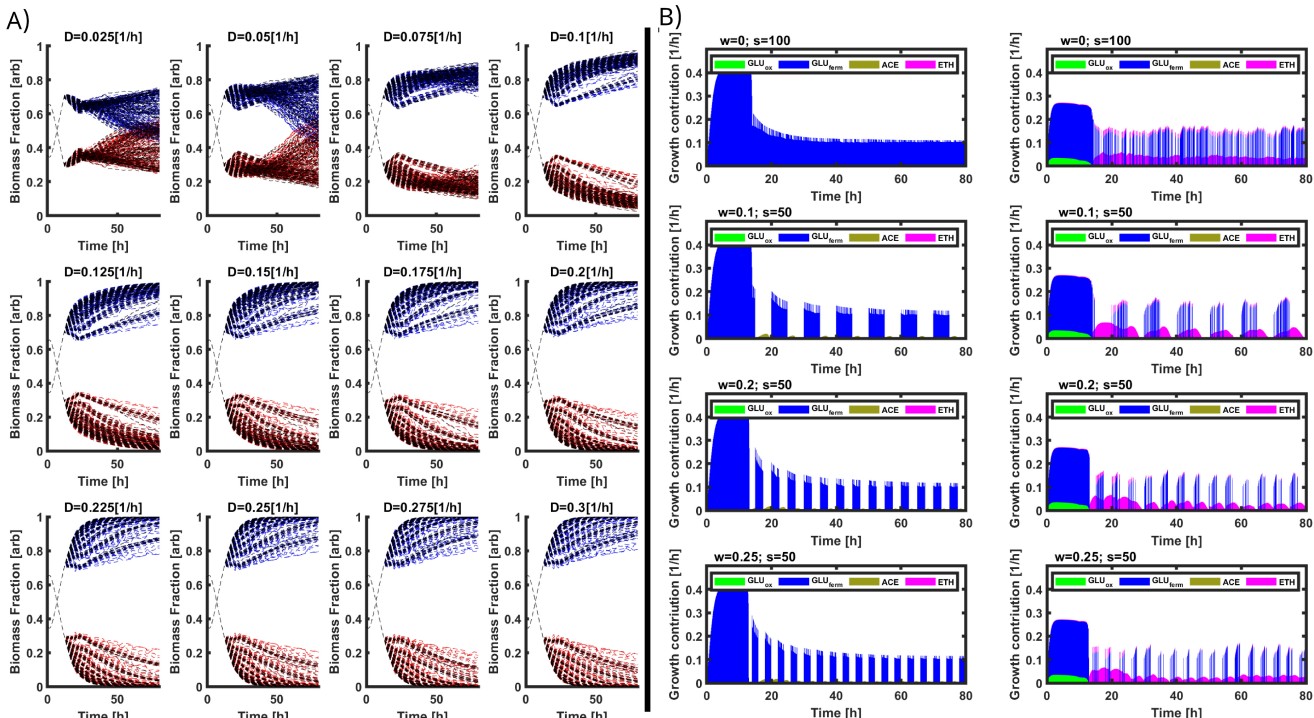

**Fig 6.** A) Simulations of the fraction for each microorganism found for the pulsed continuous cultures at different feeding profiles. E. coli (red) and S. cerevisiae (blue). B) Growth rate contributions related to each $\Psi_{rel}$ for each strain, E. coli left column, S. cerevisiae right column. Four simulations at different frequencies (w) and step-times (s) are shown. Metabolic paths are color coded as follows: GLU fermentative/unbalanced consumption in blue, ACE consumption in yellow, ETH consumption in purple, and GLU oxidative/balanced consumption in green.

for different dilution rates (**Fig 6A**) and for multiple combinations of GLU pulsing parameters (**Fig 6B**). A general feature is that *E. coli* tends to outcompete *S. cerevisiae* as *D* increases, due to the higher global growth rate observed for the bacteria.

However, the model predicts that for a range of *D* comprised between 0.025 and 0.1 h$^{-1}$, there are some GLU feeding profiles (resulting from a combination of *w* and *s* parameters) promoting the growth of the yeast and leading to the potential conservation of the species in the co-culture. We then focused on the scenario involving the highest *D* leading to the potential conservation of the species in the co-culture, i.e., *D* = 0.1 h$^{-1}$. We then computed the time evolution of growth rate for *E. coli* and *S. cerevisiae* according to different sets of GLU pulsing parameters *s* and *w* (**Fig 6B**). These simulation data clearly point out that *S. cerevisiae* can benefit from a transient fitness advantage during the periods of GLU exhaustion and when diauxic shift is occurring. Indeed, under these conditions, the yeast is able to increase its enzymatic pool related to ETH reassimilation, as shown based on the time evolution of the growth contribution of the different cybernetic variables (**Fig 6B**). Interestingly, this periodical fitness advantage was increased when the GLU pulsing frequency (*w*) increased and when the step time (*s*) decreased. The fact that cell populations can respond better to some stimulation frequencies has been previously explained from the perspective of information transmission through gene circuits [49,50], as well as from the perspective of phenotypic switching and fitness optimization [21,27]. More recently, it has also been demonstrated that the stability of microbial communities can also be strengthened by applying environmental perturbations at given frequencies [4,9,11]. In a similar way, our simulation data sets pointed out that it is

possible to ensure the stability of bacteria-yeast co-cultures by applying GLU pulsing at specific frequencies. This result will be challenged experimentally in the next section.

## Population stability resulting from intermittent feeding is frequency-dependent

Based on the simulated GLU pulsing profiles, we then run two other types of continuous culture, i.e., one at a high-frequency GLU pulsing of 0.33 h$^{-1}$ and the other at a low-frequency GLU pulsing of 0.14 h$^{-1}$. As for the chemostat experiment, these two other types of cultivation were also followed based on on-line flow cytometry (**Fig 7**). As predicted, environmental pulsing led to an oscillating profile at the level of the relative biomass fraction during co-cultivation (**Fig 7A and 7D**). This effect is promoted, after each pulse and upon GLU exhaustion, by the metabolic fitness increase of the yeast population due to diauxic shifting. This effect is clearly depicted through the time evolution of the cybernetic variables associated with ETH reassimilation (**Fig 7C and 7F**). This effect can be extended upon GLU pulsing during the whole cultivation period, i.e., approximately 80 hours. It is important to note that the population oscillations predicted by the model exhibit smaller amplitudes than the ones observed during the experiments. This difference suggests that cells in the co-culture are even more responsive than predicted by the model. The latter can be explained by the fact that the model simulates the dynamic behaviour of the co-cultures based on the individual behaviour of the microbial strains cultivated alone (**Fig 2**). Accordingly, faster changes associated with microbial social interactions, such as commensalism, predatory behaviour or symbiosis among others are not taken into account. Nevertheless, the model is very useful for determining the global behaviour of co-cultures under fluctuating environmental conditions. Probably, the most interesting feature of the model is the ability to determine the systemic property of the co-culture system to enter a state of dynamic stability, in which both microbial strains reach quasi-stable proportions and concentrations.

The results also point out that the stability of the co-culture is frequency-dependent. Indeed, for the experiment run at low-frequency, better control over the co-culture is achieved, as denoted by the regular population oscillating profile (**Fig 7A**). However, even if the fraction of yeast cells is higher in the experiment conducted with low-frequency fluctuations than in the reference chemostat (15% of yeast cells with low-frequency fluctuations, by comparison with 7% of yeast cells for the reference chemostat experiment shown at **Fig 4**, it remains below the value observed for the continuous culture with high-frequency fluctuations, i.e., 18% Fig 7B). Indeed, the continuous culture run with high-frequency perturbations exhibits a higher global biomass concentration, but also a noisier oscillatory profile, possibly impairing co-culture stability on the long run. The analysis of the simulations based on the cybernetic model points out that periodic diauxic shifts occur during the cultivation with GLU pulsing (**Fig 7C–7F**). This phenomenon has been quantified based on the relative enzyme variables, and the data points out that the yeast can beneficiate from growing on the ETH released during the main growth phase of GLU. This is particularly obvious for the cultivation carried out at low frequency pulsing (**Fig 7E**).

Cell agglomeration was observed based on the forward scatter profile of the on-line flow cytometry data (**S3 File**). This effect was quantified, but no significant differences between the three types of cultivation were observed, suggesting that this effect is independent of the feeding profile (**S3 File**). Cell aggregation has been previously observed during co-culture studies, but its origin and function are still unclear [51]. Another interesting feature is that, under GLU pulsing conditions, the model predicted more simultaneity in the occurrence of the different metabolic states based on the evolution of the cybernetic variables (**Fig 7C–7F**). This could be

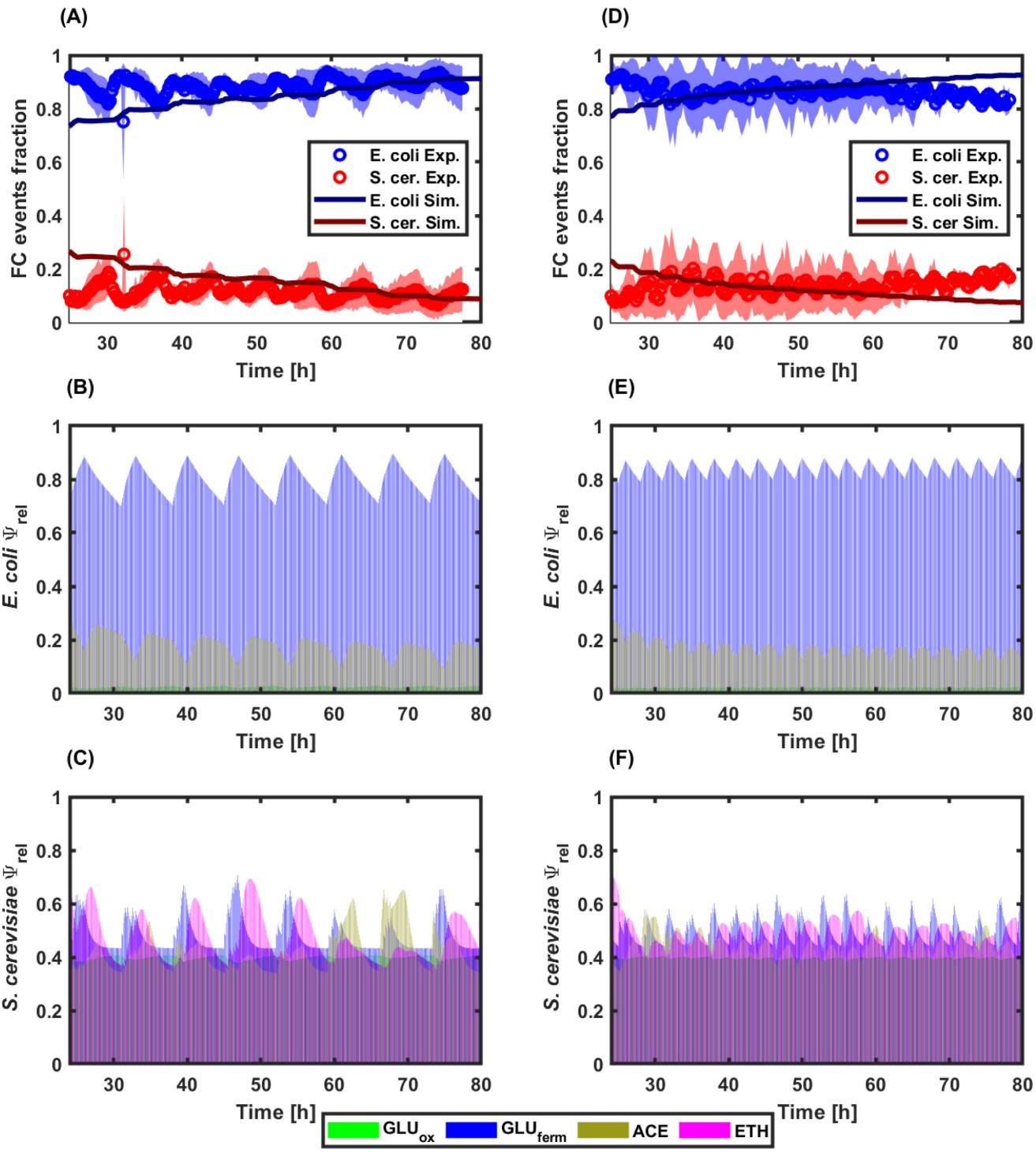

**Fig 7.** A) and D) Time evolution of the FC events fraction for E. coli (blue) and S. cerevisiae (red) at low- and high-frequency feeding profiles, respectively. B) and E) Time evolution of the relative enzyme ($\Psi_{rel}$) corresponding to each metabolic pathway for E. coli at low- and high-frequency feeding profiles, respectively. C) and F) Time evolution of the relative enzyme ($\Psi_{rel}$) corresponding to each metabolic pathway for S. cerevisiae at low- and high-frequency feeding profiles, respectively. Metabolic paths are colour coded as follows: GLU fermentative/unbalanced consumption in blue, ACE consumption in yellow, ETH consumption in purple, and GLU oxidative/balanced consumption in green.

traduced to yeast cells being able to switch quickly from one metabolic state to another, such as glycolytic to gluconeogenic, or that phenotypically different subpopulations of yeast cells are generated during the process. This hypothesis will be challenged in the next section.

## Co-culture dynamics involves the active phenotypic diversification of the yeast population

We validated the fact that successive diauxic shifts promoted the maintenance of the yeast population in the co-culture. However, the commitment to diauxic shift is known to be a mechanism exhibiting a high variability from one yeast cell to another, and this mechanism has been reported to be a major determinant of yeast fitness in carbon fluctuating environments [52,53]. Indeed, previous observations pointed out that some cells are able to switch quickly from one carbon source to another (e.g., from GLU to ETH), while other cells switch more slowly or never commit to the utilization of the alternative carbon source. This mechanism is closely related to what is often termed as bet-hedging and involves the distribution of metabolic tasks among microbial populations, e.g., by choosing between either the active investment in growth or the reduction of growth to the benefit of alternative survival strategies (use of gluconeogenic carbon sources such as ETH, accumulation of reserve carbohydrates such as glycogen) [54–56]. We then decided to use a bet-hedging fluorescent reporter to investigate the different growth/metabolic strategies exhibited by yeast cells during the continuous co-culture at low GLU pulsing frequency. In this work, we decided to use a variant of the *glc3* promoter designed to be a fluorescent reporter of the trade-off between growth and stress response. The initial *glc3* promoter was fused with HSE elements of the *hsp26* promoter to enhance the solubility of the fluorescent protein transcripts and to obtain a more reliable signal [33]. The choice of this transcriptional reporter was driven by the fact that its activation involves a growth trade-off. Indeed, a systematic analysis of the genes exhibiting an induction profile anti-correlated with the cellular growth rate has revealed a set of about 70 genes with such behaviour [55]. Among them, genes related to the synthesis of reserve carbohydrates, such as trehalose (*tsl1*) and glycogen (*glc3*), were identified. However, the correlation between the response of the bet-hedging reporter and the metabolic activities of cells is not straightforward.

In this work, we decided to consider these metabolic activities based on the use of the cybernetic model variables, i.e., relative $\Psi$ values accounting for the pool of enzymes involved in GLU respiration, GLU fermentation, ACE assimilation, and ETH assimilation, respectively. A higher $\Psi$ value means more enzymes invested in the corresponding pathway and, therefore, an increased metabolic flux through this pathway. Accordingly, we can reasonably assume that if the bet-hedging reporter shows a level of expression anti-correlated to growth, its response is also anti-correlated to the carbon flux invested in growth, directly linked with the $\Psi^{GLU}_{oxidative}$ value (**Fig 8A**, see also Eq 1 and Eq 7). What is less clear is the possible correlation between the response of the biosensor and the reassimilation of ETH and ACE.

We then challenged the response of the bet-hedging reporter in microfluidics (**Fig 8B**) and chemostat (Fig 8E). The microfluidics data pointed out that, indeed, our bet-hedging reporter exhibited anti-correlation between the level of expression and the growth (**Fig 8B and 8C and 8D**). More precisely, we observed a threshold effect when the amount of GLU perfused in the microfluidics was decreased, with a significant increase of the GFP positive fraction of cells when GLU concentration was switched from 0.15 to 0.1 mM. At this concentration range, the carbon flux is redirected from growth to an alternative direction, such as stress response [54–56]. The molecular component of this stress response was not analysed in detail, but it can be assumed that growth-arrested cells displaying a high GFP content, when cultivated in microfluidics at 0.1 and 0.05 mM, could be more adapted to exhibit diauxic shift upon consumption of alternative carbon

source. This hypothesis will be challenged later based on co-culture experiments. Additional chemostat experiments were also carried out with the reporter strain and pointed out an anti-correlation between the global growth rate (as determined based on the dilution rate $D$) and the accumulation of GFP-positive cells in the population (**Fig 8E**). Reduction in growth rate was also accompanied by an increase of the phenotypic diversity of the population.

Based on this bet-hedging reporter, co- and mono-cultures were performed based on continuous cultivation at low-frequency GLU pulsing. On-line flow cytometry allowed us to keep track of the population composition and the degree of heterogeneity for the expression of the bet-hedging reporter for the yeast. The first observation is that the phenotypic heterogeneity of the yeast cultivated alone was far below the one observed during yeast-bacteria co-cultures, as observed based on the standard deviation of the GFP distribution inside the yeast population (**Fig 9A–9D**), suggesting that microbial competition upon co-culture drives the yeast into an active diversification process. We then computed the $\Psi_{rel}$ values accounting for the pool of enzymes involved in GLU respiration, GLU fermentation, ACE assimilation, and ETH assimilation, respectively (**Fig 9E and 9F**). These values were fitted to the experimental growth rates recorded based on on-line flow cytometry (*S4 File*). Again, strong differences were observed between mono- and co-cultures. Indeed, in mono-culture, the diauxic shift exhibited sequential order with a peak in $\Psi^{GLU}_{fermentative}$ first, followed by a peak in $\Psi^{ETH}$ and, finally, a peak in $\Psi^{ACE}$ (**Fig 9E**). This succession was not observed in the case of the co-culture, where the changes associated with the enzymatic pools occurred more simultaneously (**Fig 9F**). In this case, the higher phenotypic diversity experimentally observed based on the bet-hedging reporter could be at the origin of the appearance of several subpopulations of cells with given metabolic activities. Taken altogether, the data pointed out that co-culture stability could be ensured by active diversification of the yeast population giving it more metabolic flexibility and fitness under fluctuating environmental conditions.

## Discussion

The stability of microbial co-cultures is a major line of research in systems and synthetic biology, as well as in microbial ecology. Advances in systems and synthetic biology have led to the design of synthetic genetic parts dedicated to maintaining species balance during cultivation. Most of these systems are based on cross-feeding [3,57], and quorum sensing modules [58,59], and requires the genetic engineering of the co-cultured microbial species. However, several types of cross-feeding (e.g., Substrate, metabolite, mutual, augmented cross-feeds) can be found even between wild type organisms, an thus defining their social interactions [60]. In this work, we have focused our effort on the stabilization of wild-type strains of *E. coli* and *S. cerevisiae* considering substrate competition and possible overflow metabolite cross-feeding. While these strains exhibit very different growth properties, it has been possible to ensure their stable co-culture in continuous bioreactor based on the generation of successive diauxic shifts through GLU pulsing. Based on a microbial growth model relying on a cybernetic optimization routine of Monod-type equations, we were able to predict the existence of environmental scenarios leading to population stability. For this purpose, the analysis of the cybernetic variables was crucial since it led to the generation of a transitory periodical effect allowing the yeast to coexist with the bacteria. More precisely, upon GLU exhaustion, the yeast was able to increase its metabolic fitness based on the reassimilation of the overflow metabolites ETH and ACE. Such diauxic effect has been previously reported as a key driver leading to the stabilization of microbial communities [20,61,62].

Another aspect to be considered for stabilizing the microbial co-culture is the frequency at which it is exposed to diauxic shifts. For this purpose, we applied GLU pulsing at different frequencies (i.e., low and high) to the continuous co-cultures. As predicted based on the cybernetic

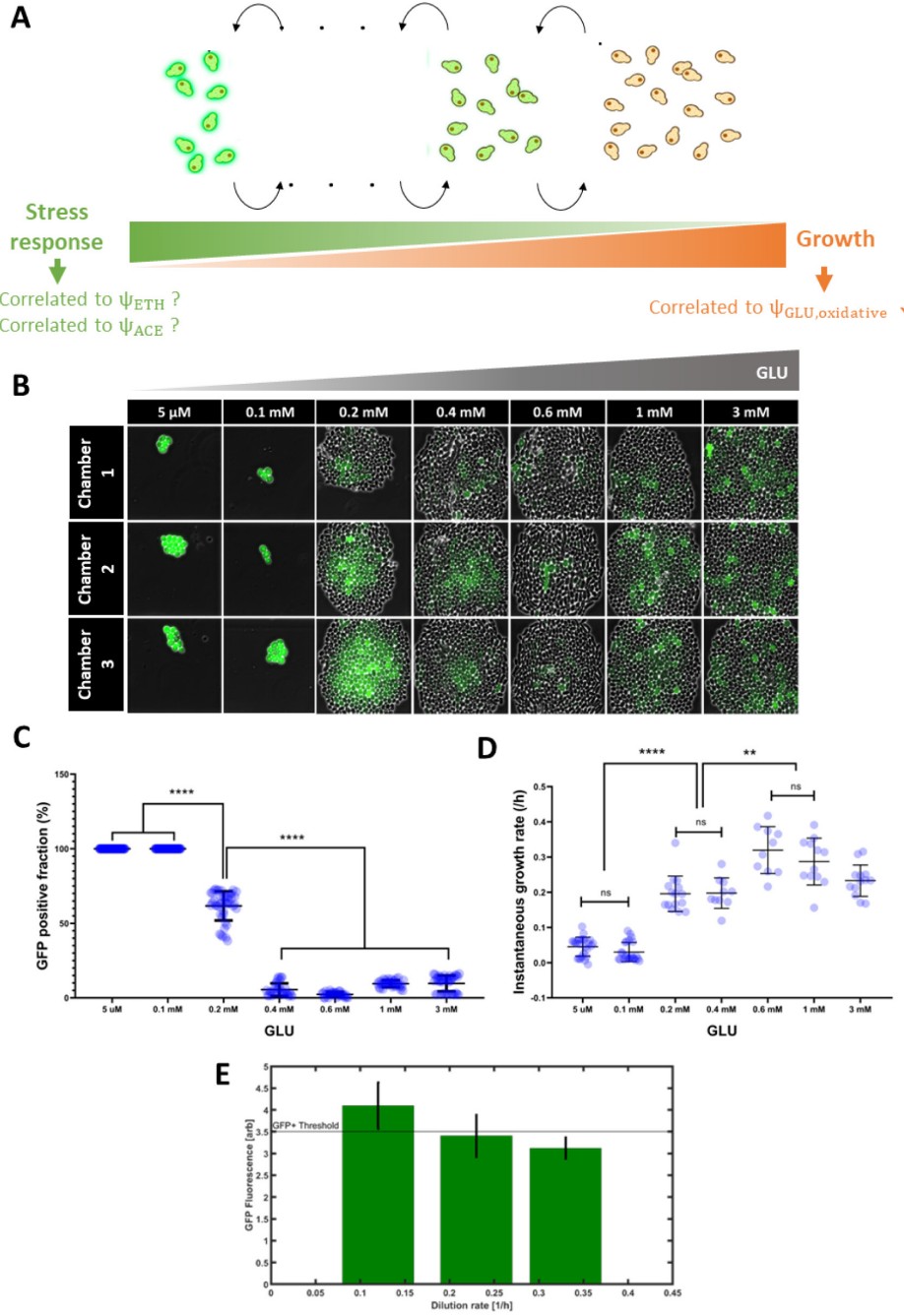

**Fig 8.** A) Scheme representing the relationship between the stress response associated to substrate limitation and the GFP fluorescence and growth rate, with several stress-to-growth intermediate states for S. cerevisiae. B) Microscopy pictures corresponding to three microfluidic chambers at different GLU feed concentrations after 48 h of cultivation. C) GFP positive cell fraction for each glucose concentration presented above (paired t-test results are shown: **** p<0.0001). D) Instantaneous growth rate, calculated from time-lapse following individual cells in the above-presented chambers, for all the GLU feed concentrations (paired t-test results are shown: **** p<0.0001; ** p< 0.01). E) relationship of the GFP fraction found in chemostat cultures at three increasing dilution rates (growth rates). The horizontal line marks the GFP positive threshold found on flow cytometry data.

 

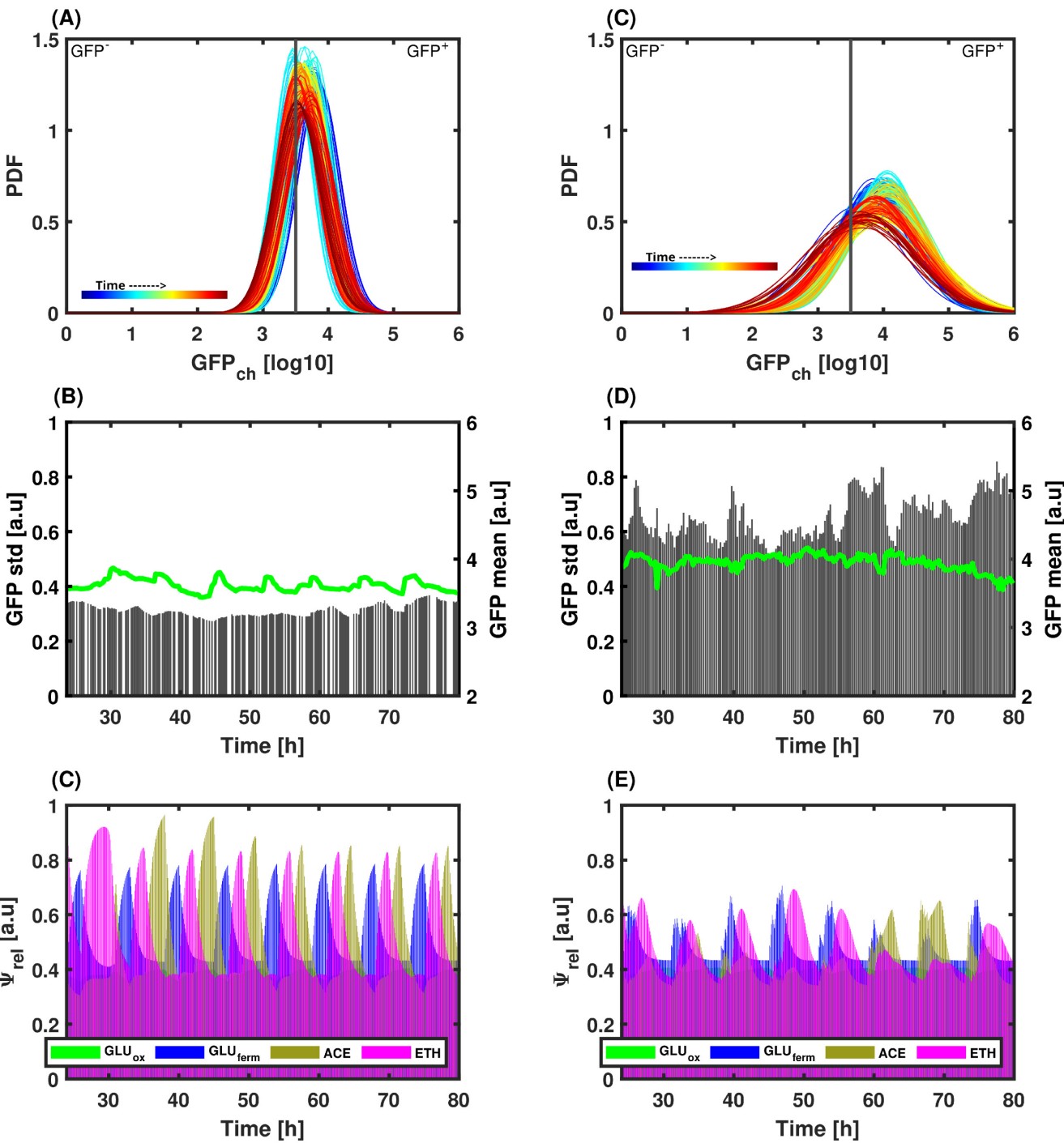

**Fig 9.** A) and C) Time evolution of the probability density functions for a low-frequency feeding regime continuous culture (A) S. cerevisiae and B) S. cerevisiae in co-culture with E. coli. Time is color-coded from blue at 24h to red at 80h. B) and D) Time evolution of the mean (green line) and standard deviation (grey bars) for the S. cerevisiae and S. cerevisiae in co-culture with E. coli, respectively. C) and E) Time evolution of the relative enzyme ($\Psi_{rel}$) corresponding to each metabolic pathway for the S. cerevisiae and S. cerevisiae in co-culture with E. coli, respectively.

 

model, both frequencies led to co-culture stabilization. However, the co-cultures run at lower GLU pulsing frequencies exhibited higher controllability on the co-cultivated strains, as deduced based on the sustained oscillations observed for the abundance of each species. This result confirms that the frequency of the environmental perturbations is a critical determinant of microbial stability, as suggested in the literature [9,11,12,63,64]. Also, biological oscillations have been previously observed in other microbial systems exposed to periodic stimulations in order to control either cell density [14] or the expression of target genes [41,65].

Generally speaking, the presence of *E. coli* in the co-culture led to a decrease in the fitness of *S. cerevisiae* during GLU utilization but to have a metabolic advantage during diauxic shifts. Indeed, the comparison of the phenotypic diversity between the mono- and co-cultures with GLU pulsing pointed out a higher heterogeneity of the yeast population when co-cultured with *E. coli*, probably due to the scarcity of carbon sources under these conditions. The latter resulted in higher flexibility for the yeast population for the consumption of ACE and ETH upon diauxic shifts. We then further analysed one possible source for this metabolic fitness and focused on the phenotypic diversity of the yeast population in mono- and co-culture. For this purpose, we used a yeast transcriptional reporter of bet-hedging, i.e., the cellular decision-making process driving the yeast into an actively growing state or to alternative states exhibiting reduced growth but enhanced survival. The heterogeneity in growth and stress response capability of the yeast population seemed to be at the origin of the higher fitness upon diauxic shifts. Indeed, our data point out that, upon GLU pulse, the phenotypic diversity of the yeast population increases as a direct consequence of the competition for the substrate with the bacteria. Since the role of phenotypic diversification on co-culture stability is still unclear with contradictory results found in the literature [66–68], our approach could be used in the future in order to investigate more in detail the relevance of such mechanisms.

Besides the technical challenges associated with the stabilization of microbial co-culture in continuous bioreactor and the more fundamental aspects related to the impact of phenotypic diversification, one other interesting point of this work is the possible generalization of our findings to other species. For this purpose, we developed a modelling toolbox called MONCKS, allowing us to assemble multiple ODEs-based metabolic models accounting for individual microbial species. The only requirement for the application of MONCKS to the simulation of microbial co-culture is the determination of the main metabolic pathways involved in the assimilation of the primary substrates used for the growth of the individual species (i.e., in our case GLU, but also all the side metabolites released upon overflow metabolism). This metabolic pathway determination can be performed by product yield characterization on single substrate growth experiments or by yield analysis from stoichiometric matrices reconstructed from partial (e.g., core-metabolism) or genome-scale annotations [24,30,31,69,70]. It is relevant to state that the model can be extended within the same framework to address more complex metabolite functions, internal fluxes, specific inhibitions, and interactions using the same cybernetic modelling approach. Therefore, the presented framework could be extended and refined with further experimental data to account for different characteristics of specific microbial interactions. The model presented contributes towards the construction of modular frameworks and virtual co-culture strains databases to understand, design and control microorganism populations during continuous bioprocesses.

## Supporting information

**S1 File. Extended model description.** File containing an extended description of model construction and parametrization.
(PDF)

**S2 File. Simplified MONCKS simulation workflow.** File containing a MONCKS usage and workflow description.
(PDF)

**S3 File. Simplified mFCA FC data treatment and workflow.** File containing a description of the FC data treatment methods and the use and simplified workflow on the mFCAtoolbox. Figure containing the time evolution of aggregates for S. cerevisiae and E. coli based on FC data analysis and a Figure containing the FSC-A vs. event measured fraction plots and micrography samples, respectively, at 80 h for the continuous and high-frequency feeding regime.
(PDF)

**S4 File. Continuous and Discontinuous Co-culture data.** File containing extended data for the metabolite experimental data and model simulations for continuous and the high- and low-frequency feeding regimes. Extended figures related to the S. cerevisiae relative enzyme for monoculture and co-culture at low-frequency feeding regimes.
(PDF)

**S5 File. Nomenclature.** List of abbreviations used in this work.
(PDF)

## Acknowledgments

We would like to thank Vincent Vandenbroucke for his help reviewing this manuscript.

## Author Contributions

**Conceptualization:** J. Andres Martinez, Frank Delvigne.

**Data curation:** Frank Delvigne.

**Formal analysis:** J. Andres Martinez, Frank Delvigne.

**Funding acquisition:** Christian Dusny, Frank Delvigne.

**Investigation:** J. Andres Martinez, Matheo Delvenne, Lucas Henrion, Fabian Moreno, Samuel Telek.

**Methodology:** Lucas Henrion, Fabian Moreno, Frank Delvigne.

**Project administration:** Frank Delvigne.

**Supervision:** Frank Delvigne.

**Visualization:** Matheo Delvenne.

**Writing – original draft:** Frank Delvigne.

**Writing – review & editing:** J. Andres Martinez, Christian Dusny, Frank Delvigne.

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
