## [Decision Letter · Decision Letter 0]

9 Sep 2022

Dear Dr Delvigne,

Thank you very much for submitting your manuscript "Controlling microbial co-culture based on substrate pulsing can lead to stability through differential fitness advantages" for consideration at PLOS Computational Biology. As with all papers reviewed by the journal, your manuscript was reviewed by members of the editorial board and by several independent reviewers. The reviewers appreciated the attention to an important topic. Based on the reviews, we are likely to consider this manuscript favorably, providing that you modify the manuscript according to the review recommendations.

Sincerely,

Sunil Laxman, PhD

Guest Editor

PLOS Computational Biology

Mark Alber

Section Editor

PLOS Computational Biology

[LINK]

Reviewer's Responses to Questions

**Comments to the Authors:**

Reviewer #1: In the present manuscript Martinez et al utilized a cybernetic model to study the co-culture of two classic model organisms, Escherichia coli and the Saccharomyces cerevisiae. The authors acquired extensive experimental data first by growing the microbes in monoculture and subsequently in co-culture with different glucose pulses. These experiments offer a promising proof of concept for their simulations as well as reveal additional information on the social interactions of these microbes. The described workflow is interesting and holds value for numerous applications. However, there are some open issues and addressing them will improve the clarity of their findings.

Major comments

1. Line 294 “The beginning of the culture and remains as such as long as the GLU concentration is non-limiting (> 1.5 g/L).”. Could the authors justify how they selected this value? They do not reference any of their figures and when Fig. 2A is carefully considered growth of E. coli seems limited with 2.5 g/L of glucose and severely impaired even with 5 g/L.

2. Line 506 “Most of these systems are based on cross-feeding and quorum sensing modules”. In this part of the discussion the authors aim to separate their approach from other approaches based on cross-feeding. However, their co-culture remains stable based on the consumption of “metabolic by-products” like acetate. I would argue that S. cerevisiae taking up and consuming the acetate that was produced from E. coli is a form of metabolite cross-feeding as described in Smith et al. 2019 (https://doi.org/10.3389/fevo.2019.00153). Could the authors elaborate how they believe that their tool is distinct from other modelling cross-feeding based approaches?

3. Line 554 “The only requirement for the application of MONCS to the simulation of microbial co-culture is the determination of the main metabolic pathways involved…”. In this study the authors chose very well characterised model organisms and still had to contact long and sophisticated experiments in mono-culture prior to simulating the co-cultures. Could they describe an example how they envision use of their tool, as this above described statement seems like an oversimplification for the required model input.

Minor comments

1. For figure 2, inclusion of the legend and the explanation of the symbols and colours on the figure and not only on the legend, would increase the clarity of the whole figure.

2. Figure 6 i(especially 6B) is illegible fact that limits the comprehension of the depicted data.

3. Clarity of Figure 9A and 9C would improve with inclusion of a grade scale for the colours.

Reviewer #2: The manuscript is much improved – I found it very easy to follow the experimental design and the authors did a good job of connecting each section of the results. I have a couple of comments that I am sure will be easy for the authors to address.

The authors use biomass is the measure of the amount of each of the species. An important difference between E. coli and Yeast, is there a size difference, with yeast being ~10 times bigger than E. coli. This is important when taking dilution into account for experiments- not be a problem for the range that the authors show in figure 6 but perhaps they could note that for extreme dilutions, say, in the range of 0.001 or more, it may be that Yeast are in more danger of being diluted from the culture (and therefore, to go extinct) than E. coli. Did the authors explore the greater level of dilution and its affect on the co-culture? The authors may want to briefly mention what the effect of a greater dilution would be, since one might assume from those simulations that more dilution would lead to more stability.

The figures look really great and the authors should be commended on putting together such a clear illustrations that also help the reader understand the manuscript. It would be helpful if the font sizes for figures 5, 6, and 8 etc, be standardised to a larger size since they are currently difficult to read.

Reviewer #3: In the manuscript titled ‘Controlling microbial co-culture based on substrate pulsing can lead to stability through differential fitness advantages’, the authors develop a cybernetic model to understand the dynamics in microbial co-culture. To illustrate this, the authors perform the experiments and simulations using two model organisms S. cerevisiae and E. coli. While these organisms and their cocultures are very well-studied in literature, the authors here seek use cybernetic modelling to demonstrate its’ applicability. Overall, I find the work quite interesting, however have a lot of concerns that need attention before the manuscript can be considered for publication. I have divided my comments based on different sections.

Materials and methods [Main text and Supplementary Files]

Please explain the meaning of all the variables that have been used in different equations. It is very hard to interpret without them.

Cybernetic modeling reveals differences in metabolic fitness upon the diauxic shift in mono-cultures

The authors seem to use monoculture values to model coculture.

Since the authors’ main theme is to perform cocultures, I’m not too sure if the parameters obtained from monocultures can be used for coculture modelling.

The authors also do not include an interaction term between E. coli and S. cerevisiae in the cybernetic model, which would have effects on the predictions [as authors themselves recognize in Lines 392-398]. One possible way to account for this might be to do a coculture experiment and determine the parameters to fit into the model.

Fitness disparity between metabolic phenotypes leads to population imbalances during continuous cultures with constant environmental conditions

1. Does ethanol production by S. cerevisiae inhibit the growth of E. coli? Was this factor included in the model?

2. Figure 4A – What is meant by fraction of events?

3. Figure 4A & B – It seems a little odd that E. coli psi values are constant throughout the simulations. I would have imagined these values to show a trend similar to Figure 4C [GLUferm].

4. Figure 4C- it is very hard to see the trend for ACE. Suggest the authors to plot these values as separate graph.

Fluctuating environmental conditions provide periodic fitness advantage and can lead to population stability in continuous cultures

Lines 354-374, Figures 6A, 6B

1. The authors conclude from Figure 6A that for a few values of D [ranging from 0.025-0.1 h-1], the coculture is stable. However, looking at the figures, it seems to me that E. coli is growing at the cost of S. cerevisiae at D >= 0.075 h-1, similarly the converse for D <0.075h-1 and that the co-culture is not ‘stable’ per se.

2. Figure 6B & Lines 354-374 – The Figure 6B shows all the growth rate simulations at w = 0.1, s= 0.1 and are discordant with Lines 364-366

3. Line 365 – Fitness advantage – How do the authors define fitness advantage? Is it the ability of the organism to grow at its’ maximum capacity?

Population stability resulting from intermittent feeding is frequency

1. Figure 7A – What is meant by fraction of events?

2. Lines 394-398 – Suggest the authors to include interaction term (as above).

3. Figure 7F – Same comments as Figure 4C.

Minor comments

1. Page14 – Line 355 – What are w and s parameters? Please mention in the main text

Co-culture dynamics involves the active phenotypic diversification of the yeast population

Authors carry out experiments with GFP tagged S. cerevisiae and seek to show that there is a variation in phenotype. Following points are less clear to me:

1. How can the same set of parameters determined with wild type S. cerevisiae be used for GFP tagged S. cerevisiae? I would imagine these to be different for both.

2. Figure 9 – There are two (C), please modify the names.

3. Line 496 – Line 501 – Without E. coli data, it is hard to conclude Lines 498-501.

**Have the authors made all data and (if applicable) computational code underlying the findings in their manuscript fully available?**

Reviewer #1: Yes

Reviewer #2: **No: **In supplementary data file 2, the authors provide details on the simulations however this file needs to be updated with the correct information and links so that the readers can access MONKS, this is currently not available and there are placeholder symbols where links should be

Reviewer #3: Yes

PLOS authors have the option to publish the peer review history of their article (what does this mean?). If published, this will include your full peer review and any attached files.

Reviewer #1: No

Reviewer #2: No

Reviewer #3: No

Figure Files:

Data Requirements:

Reproducibility:

References:

---

## [Decision Letter · Decision Letter 1]

22 Oct 2022

Dear Dr Delvigne,

We are pleased to inform you that your manuscript 'Controlling microbial co-culture based on substrate pulsing can lead to stability through differential fitness advantages' has been provisionally accepted for publication in PLOS Computational Biology.

Best regards,

Sunil Laxman, PhD

Guest Editor

PLOS Computational Biology

Mark Alber

Section Editor

PLOS Computational Biology

---

## [Editor Report · Acceptance letter]

25 Oct 2022

PCOMPBIOL-D-22-01232R1 

Controlling microbial co-culture based on substrate pulsing can lead to stability through differential fitness advantages

Dear Dr Delvigne,

I am pleased to inform you that your manuscript has been formally accepted for publication in PLOS Computational Biology. Your manuscript is now with our production department and you will be notified of the publication date in due course.

With kind regards,

Zsofia Freund
